# Distinct cellular states determine calcium signaling response

Jason Yao, Anna Pilko & Roy Wollman[*] iD

## Abstract

The heterogeneity in mammalian cells signaling response is largely a result of pre-existing cell-to-cell variability. It is unknown whether cell-to-cell variability rises from biochemical stochastic fluctuations or distinct cellular states. Here, we utilize calcium response to adenosine trisphosphate as a model for investigating the structure of heterogeneity within a population of cells and analyze whether distinct cellular response states coexist. We use a functional definition of cellular state that is based on a mechanistic dynamical systems model of calcium signaling. Using Bayesian parameter inference, we obtain high confidence parameter value distributions for several hundred cells, each fitted individually. Clustering the inferred parameter distributions revealed three major distinct cellular states within the population. The existence of distinct cellular states raises the possibility that the observed variability in response is a result of structured heterogeneity between cells. The inferred parameter distribution predicts, and experiments confirm that variability in IP3R response explains the majority of calcium heterogeneity. Our work shows how mechanistic models and single-cell parameter fitting can uncover hidden population structure and demonstrate the need for parameter inference at the single-cell level.

**Keywords** calcium signaling; cell states; cellular heterogeneity; single-cell biology
**Subject Categories** Quantitative Biology & Dynamical Systems; Signal Transduction
**Mol Syst Biol. (2016) 12: 894**

## Introduction

Cell-to-cell variability in dynamic responses to stimuli is observed ubiquitously (Geva-Zatorsky *et al*, 2006; Cohen-Saidon *et al*, 2009; Tay *et al*, 2010; Selimkhanov *et al*, 2014), yet its underlying causes are still unknown. Previous work decomposed biological processes in gene expression and signal transduction into intrinsic fluctuations and pre-existing variability (Elowitz *et al*, 2002; Rhee *et al*, 2014; Selimkhanov *et al*, 2014). Specific measurement of the relative contribution of stochastic fluctuations during signal transduction compared to cell-to-cell variability showed that 50–98% of the variability could be attributed to pre-existing cell-to-cell variability (Toettcher *et al*, 2013; Selimkhanov *et al*, 2014). While cell-to-cell variability was determined to be the dominant source of heterogeneity in cellular response, the cause of cell-to-cell variability is still unknown. Two hypotheses can explain the observed cell-to-cell variability: The first is that variability is a result of accumulation of stochastic fluctuations in gene expression (Shibata & Fujimoto, 2005; Ladbury & Arold, 2012; Rhee *et al*, 2014), organelle composition (Oates, 2011), and other cellular factors where small numbers of molecules increase biochemical randomness. Under this hypothesis, a clonal population will accumulate changes over time that would limit biochemical reaction accuracy. The second hypothesis is that the population of cells contains several deterministic cellular states, according to the surrounding microenvironment, or due to functional differences between cells, in which case cells would react in predictable patterns of behaviors (Snijder *et al*, 2009). While these two hypotheses seem contradictory, they are not mutually exclusive and each of these hypotheses can explain a different amount of the total observed variance. One approach that was used to test the different contributions of these two mechanisms is to predict the observed response variability based on measurements of large arrays of features for each cell (Snijder *et al*, 2009). However, it is possible that the measured features themselves include information that is indicative of the accumulation of the stochastic fluctuation. Therefore, while the measured features have high predictive power, it does not provide an indication of distinct functional states. Another approach to identify functional cell state is based on a dynamical systems point of view, which argues that a cell can be represented as a dynamical system and that each cell state is an attractor within the cell state space (Furusawa & Kaneko, 2012). The hallmark of this multiple-attractor view is the existence of distinct clusters within cell state space. While distinct clusters are indicative of multiple distinct cell states, not every functional heterogeneity will show distinct clusters. It is possible that because cells need to comply with multiple objectives, they occupy a Pareto front and therefore functional heterogeneity will not necessarily cause distinct cell states (Sheftel *et al*, 2013). Nonetheless, the existence of distinct cell states can provide support for the existence of attractors and therefore the possibility that structured heterogeneity contributes to the observed cellular variability.

Departments of Chemistry and Biochemistry, Integrative Biology and Physiology, and Institute for Quantitative and Computational Biosciences (QCB), UCLA, Los Angeles, CA, USA
*Corresponding author. Tel: +1 858 210 0905; E-mail: rwollman@ucsd.edu

The true complexity of cellular state space is unknown and can only be probed by examining specific signatures that indicate the underlying cell state. There are numerous forms for possible signatures, such as transcript abundance and protein concentrations. Many of these signatures have thousands of dimensions, and therefore, dimensionality reduction approaches are routinely used to reduce these large spaces into fewer dimensions (Buettner *et al*, 2015). However, statistical methods on their own do not provide any mechanistic insights toward the meaning of any identified clusters. An additional limitation with statistical dimensionality reduction approaches is that they do not take into account the substantial body of prior knowledge on the underlying biochemical mechanisms which contribute to the variation. The kinetic parameters of the reactions within the signaling network are an alternative and natural cell state signature. Kinetic parameters are aggregates of molecular activities from the genetic level upwards that provide a functional representation of cell-to-cell variability in the dynamics of signal transduction pathways. By employing differential equation modeling, we can simulate the dynamic output of each signature and connect the underlying state variability with the observed response variability. Therefore, kinetic parameters could provide a useful signature of cellular state that could be used to determine the underlying distribution of cell heterogeneity.

Determining cellular kinetic parameters is challenging, because experimental measurement of kinetic parameters of biochemical reactions *in vivo* with high accuracy is extremely difficult. An alternative approach is to infer the kinetic parameter values from cellular responses (Lee *et al*, 2009). Parameter inference allows for the learning of parameter values for a differential equations model of signal transduction based on measured time-series data of state variables in the model (Eydgahi *et al*, 2013). Parameter inference of signal transduction networks is a difficult problem due to the large number of state variables and parameter values which would create non-identifiability issues (Raue *et al*, 2011) or challenges from model "sloppiness" where multiple sets of parameters could fit the data equally well (Gutenkunst *et al*, 2007). Thus, to utilize differential equation modeling to make meaningful inference about hidden cell states, it is important to account for the uncertainty associated with parameter inference. This could be done through the use of Bayesian inference to infer the full posterior parameter distribution for each parameter. The full distributions provide information on the accuracy of the inference and report on existence of any "sloppy" eigenvectors of parameters within goodness-of-fit space.

Through the use of full distributions of kinetic parameters as cellular signatures, it is possible to glean insights into cellular state distribution. Here, we inferred cell states of signaling dynamics by fitting a population of single-cell calcium trajectories to a mathematical model and produced a distribution of parameter ensembles. We first measured cellular calcium responses to adenosine trisphosphate (ATP) stimulations using fluorescence microscopy and MATLAB image processing to obtain single-cell calcium responses. The single-cell data were fitted to a system of differential equations of the calcium signaling pathway using Bayesian inference to obtain the posterior distribution of parameter values vectors within a certain level of tolerance for each of the fitted trajectories. We analyzed the distribution of parameter ensembles using the Kullback–Leibler divergence criteria and found three major distinct clusters of parameter ensembles. The identification of major clusters indicates the possibility that cell-to-cell variability could be interpreted not as a simple variability that stems from accumulation of stochastic fluctuations but rather from the existence of multiple distinct states within the population.

## Results

We measured calcium responses of a non-tumorigenic human mammary epithelial cell line (MCF10A) to stimulation with 10 μM ATP using the Fluo-4 calcium indicator calibrated by using an established protocol (Bao *et al*, 2010). The fluorescence images were segmented, and cells were identified and tracked over time to measure single-cell dynamic calcium responses. Figure 1 shows the response matrix (Fig 1A), population average (Fig 1B) and a few representative examples (Fig 1C) of single-cell calcium responses to ATP. As was previously reported for this system (Selimkhanov *et al*, 2014), cellular responses were highly heterogeneous. We analyzed the following response dynamics: response in contrast with basal level before stimulation, latency to reach half maximal level, latency to reach maximum level, maximum level, time from maximal level to half maximal, and steady state after stimulation (Fig 1D) (Cohen-Saidon *et al*, 2009). The results showed wide heterogeneity with respect to the trajectory features (Fig 1E). The distribution of cells with similar trajectory features were not spatially correlated (Fig 1E and F). We initially attempted to cluster the single-cell calcium trajectories based on either the entire calcium responses or through their representative time-series features. The obtained clusters did not identify clear response archetypes (Fig EV1) potentially as a result of the fact that the ratio of variance between clusters to the intracellular variance was low (Fig EV1D). Therefore, while clustering based on time-series data is a valid and useful approach, in this case it did not lead to identification of cell states. To further explore the possibility that cell states exist, we hypothesize that insights regarding cellular states could be uncovered by studying the mechanistic structure of pathway using mathematical modeling that incorporate prior knowledge on the underlying signaling network.

We used the mathematical model of calcium signaling from Lemon *et al* (2003) and Li and Rinzel (1994) as a template to construct a model of the calcium signaling response. The model was simplified to improve parameter identifiability. Overall, the mathematical model included 17 parameters and four state variables. Several simplifications and assumptions were included as follows: there is no mitochondrion; the model does not include plasma membrane leakage or store operated channel for calcium therefore total cellular calcium is kept constant. While the model does not capture all the known biochemical details, it is simplified to attenuate model redundancy and increase model identifiability. The schematic diagram of calcium signaling is shown in Fig 2. According to our model, when the ligand ATP binds to the purinergic receptor, the binding action triggers a series of events leading to the activation of enzyme phospholipase C (PLC). PLC then hydrolyzes phosphatidylinositol 4,5-bisphosphate (PIP2) to produce inositol 1,4,5-trisphosphate (IP3) and diacylglycerol (DAG). IP3 activates the calcium channels on the endoplasmic reticulum (ER), which would release calcium from ER into the cytosol, producing a calcium spike. The sarco/endoplasmic reticulum $Ca^{2+}$–ATPase (SERCA) channel on the ER then pumps the calcium in the cytosol back into the ER, thus completing the transient calcium spike (Lemon *et al*, 2003).

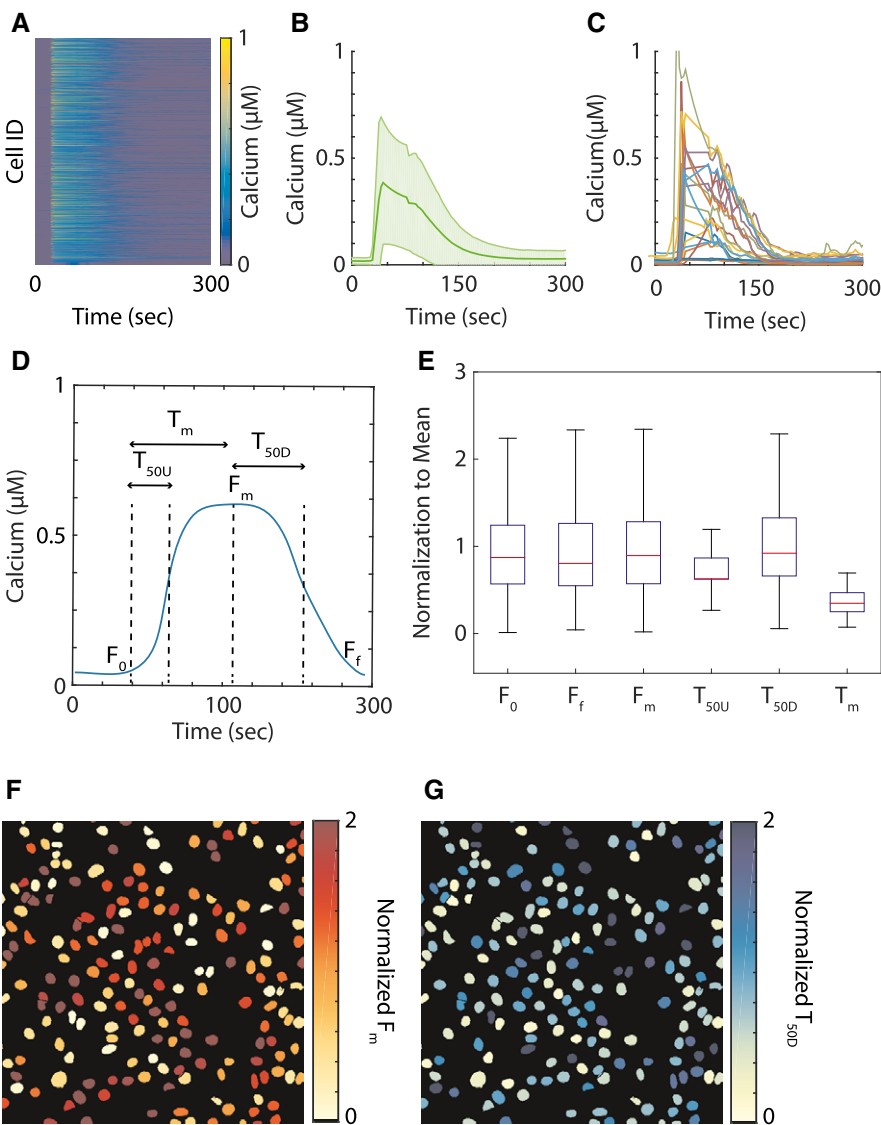

**Figure 1.  Calcium signal response heterogeneity.**

A     A matrix of single-cell calcium response to ATP perturbation. Each row represents the calcium response of a single cell, and columns are time points. The heatmap figure contains data from a total of 1,904 cells.

B     The average calcium response of the population. The solid green line is the population average, and the light green shade is the standard deviation.

C     Examples of single-cell calcium data.

D     Example of single-cell calcium data with representative features of the basal value ($F_0$), time to reach half maximum ($T_{50U}$), time to reach maximum ($T_m$), maximum value ($F_m$), time to decay to half maximum ($T_{50D}$), and steady-state final value ($F_f$).

E     Boxplots of representative time-series features of calcium signals in the cell population, normalized to the average value of the particular feature. The red lines of the boxplots represent the median of the normalized values, and the ranges of the whiskers are defined to be 1.5 times the interquartile range (difference between 75[th] and 25[th] percentiles).

F, G   Spatial distributions within one representative well of the maximum signal ($F_m$) and the time to decay to half maximum ($T_{50D}$).

Source data are available online for this figure.

To uncover the functional signatures of cell states, we inferred the kinetic parameter distributions for each of the single-cell trajectories using a Bayesian algorithm. The entire workflow, including raw data signal processing, data fitting, convergence testing, and sampling parameter distributions, is outlined in Fig 3A. Initially, the data were processed to remove experimental noise and other high-frequency elements of calcium response that were not captured by the model. We found that pre-processing the data increased the quality of the identified model fits, likely because it effectively reduced the stochastic and technical noise. The square of the sum of difference between the processed and raw time-course calcium data was later used to determine the rejection cutoff of a plausible model. For parameter inference, we used a variant of approximate Bayesian computation-based sequential Monte Carlo (ABC-SMC) method for

our single-cell trajectory fitting (Liepe *et al*, 2014). In the Bayesian framework, we inferred parameter values for the system of differential equations in terms of their posterior distributions (Liepe *et al*, 2014) corresponding to single-cell calcium responses. At the core of our method is the ABC rejection algorithm, which arrives at the posterior distribution through simulating the differential equation model, calculating the agreement between simulation results and the experimental data and rejecting parameter values that have low agreement with the data. The key to efficient ABC sampling is using a good proposal distribution such that the fraction of rejection parameters is low. The ABC-SMC algorithm iteratively updates the proposal distribution through repeated ABC sampling where a proposal distribution (i.e., prior) of one iteration is based on the posterior of the previous iteration. In each step, the proposal distribution and the rejection cutoff are updated to balance the need to explore unseen regions of parameter space with the need to exploit regions that provide good fits. The iterations continue until the parameter distribution converges to within the experimental error that is determined through low-pass filtering of the data. During sampling iterations, the region that the parameter vectors occupy converges. An example of the projection of the parameter distributions from multiple iterations on 2D space is shown in Fig 3B. The parameters were sampled in log scale with the initial prior distribution ranging over two orders of magnitude and centered on the referenced values that were chosen based either on the literature or our estimates of biologically reasonable ranges.

A key challenge in any optimization in a large parameter space is the tendency to identify only local solutions. This is especially a concern due to the probabilistic nature of our search algorithm. To verify that our search algorithm samples correctly from the target distribution, we used strict convergence criteria. We tested for convergence by fitting each of the single-cell responses twice. This means that one single-cell trajectory would have two populations of parameter vectors run #1 and run #2 which represented the posterior parameter distributions associated with the two independent runs of the algorithm. We reasoned that if the two posterior populations were from the same distribution, then the nearest neighbors of the parameter vectors from those two runs should come from either population with equal probability. This was the null hypothesis of our test. To test whether the nearest neighbor probability that we calculated shows a statistically significant difference from the null hypothesis of equal probability, we used a chi-square test. The two runs were considered to be convergent if the test failed to reject the null hypothesis. To demonstrate the overall parameter convergence, Fig 3C shows the multidimensional scaling projection onto a 2D plane from two sets of fitting to trajectories, one is convergent and the other non-convergent. Note that the 2D projection is presented for visualization purposes only and the test was conducted in a 17-dimension parameter space. The result of the algorithm for a single trajectory produced a population of parameter vectors representing the individual parameters in terms of their confidence intervals (Fig 3D). We successfully identified 672 cellular responses with corresponding convergent parameter distributions (Fig EV2).

To determine whether distinct cell states exist, we clustered the identified parameter distributions for each of the cells (Fig 4). The key step in any cluster analysis is to determine a distance between any two cells in the cell state space. To take into account the

**A**

$$\frac{d[PLC]}{dt} = K_{on,ATP} * [ATP] * e^{-K_{ATP}t} - K_{off,PLC}[PLC]$$

$$\frac{d[IP3]}{dt} = V_{PLC} \frac{[PLC]^2}{K_{IP3}^2 + [PLC]^2} - K_{off,IP3}[IP3]$$

$$\frac{dh}{dt} = a\left([Ca^{2+}] + d_{inh}\right)\left(\frac{d_{inh}}{[Ca^{2+}] + d_{inh}} - h\right)$$

$$\frac{d[Ca^{2+}]}{dt} = \beta\left(\varepsilon(\eta_1.m_\infty{}^3.h^3 + \eta_2)(c_0 - (1+\varepsilon)[Ca^{2+}]) - \eta_3\frac{[Ca^{2+}]^2}{k_3{}^2 + [Ca^{2+}]^2}\right)$$

$$\beta = \left(1 + \frac{K_e[B_e]}{(K_e + [Ca^{2+}])^2}\right)^{-1} \qquad m_\infty = \left(\frac{[IP3]}{d_1 + [IP3]}\right)\left(\frac{[Ca^{2+}]}{d_5 + [Ca^{2+}]}\right)$$

**B**

**Figure 2. Calcium model.**

A   A system of ordinary differential equations of calcium signaling. The system contains four state variables of IP3, PLC, calcium, and IP3-activated receptor fraction.

B   Schematic of calcium signaling pathway. Through ATP stimulation, GPCR is recruited to activate enzyme PLC, which in turn cleaves PIP2 to form IP3 and DAG; IP3 then activates the IP3R channel to release calcium into the cytoplasm. The calcium is recycled back into the ER membrane through the SERCA pump.

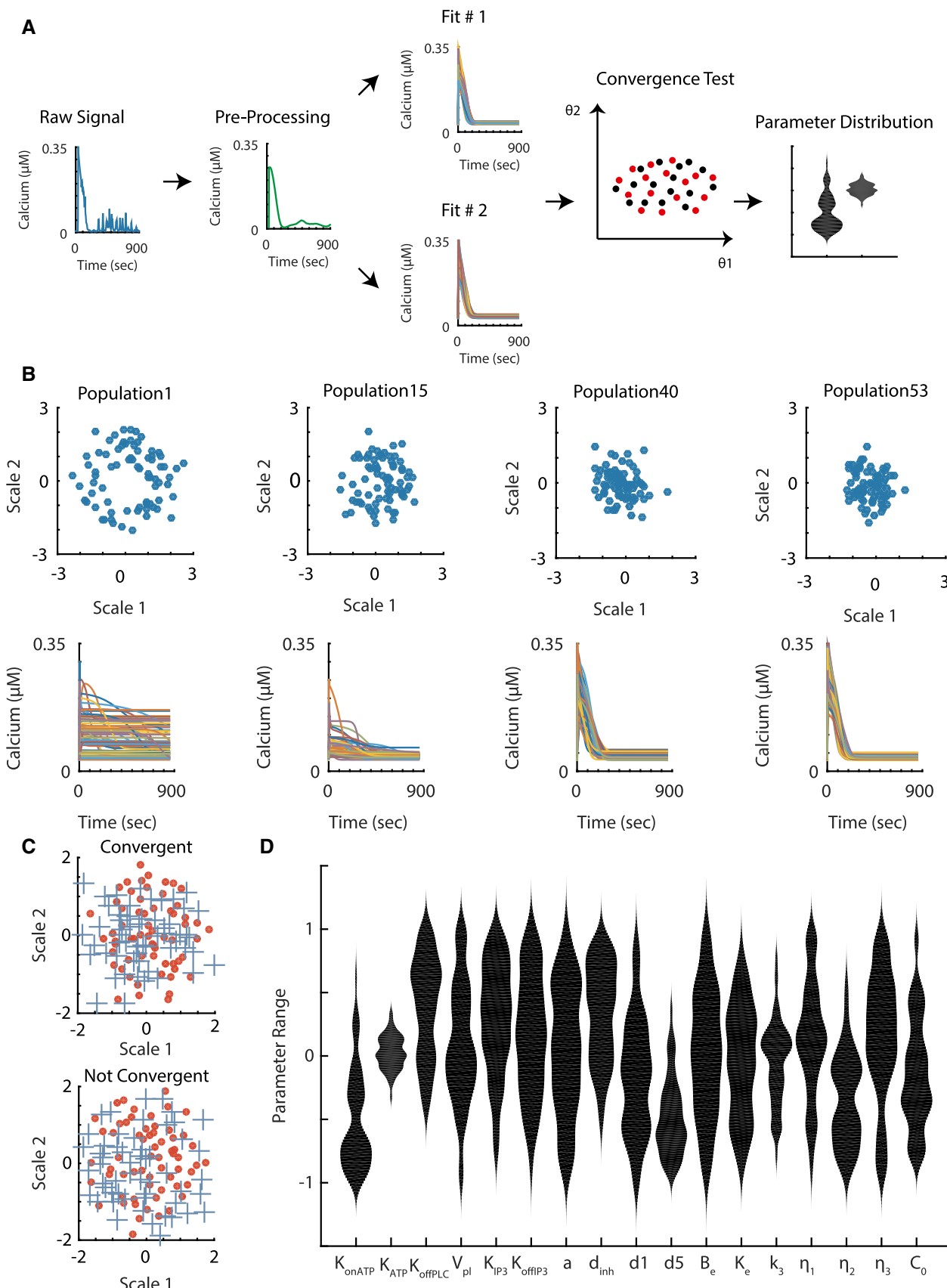

Figure 3.

**Figure 3.  Process of single-cell parameter fitting.**

A   Workflow of the single-cell data fitting process. The raw data are pre-processed to remove the technical noise and high-frequency biological responses. The model is fitted to the processed data in two independent runs of the algorithm to generate two posterior parameter samples. A statistical test based on similarities of the two independent runs was used to determine if the algorithm successfully converges to same posterior distribution. The result of the entire workflow is a population of parameter vectors that represent the single-cell fitting.

B   The progression of parameter fitting for a single trajectory. The parameter vectors through the iterations were projected using multidimensional scaling onto a 2D plane to illustrate the convergence of the parameters. The simulated data for each of the selected iterations are plotted below.

C   Comparison between a convergent runs and non-convergent runs. The crosses and the filled circles represent independent runs of the same single-cell data. The parameter posterior is projected onto the 2D plane through multidimensional scaling. There is a wider separation of parameter groups in the non-convergent fitting as opposed to the convergent run.

D   The violin plot of all kinetic parameters from the fit results to a single-cell calcium trajectory. The plot values are in terms of the log of 10 of the parameter values normalized according to the reference values determined either through literature or values within reasonable biological ranges. The meanings of the kinetic parameter symbols and their reference values are listed in Table EV1.

Source data are available online for this figure.

multidimensional confidence intervals provided by the parameter distribution, we used a Kullback–Leibler (KL) divergence measure, an information theoretic measure that determines the similarity between parameter ensembles. For every two cells, we constructed a probability distribution of the likelihood of nearest neighbor parameter vector to be from the same cells. The distance measure was defined as the KL divergence between the constructed distribution and a null model where the parameter vector sets for the two cells are fully mixed. Combining the full probability distribution and the use of KL divergence resulted in a distance measure that integrates information on the difference between parameter values and on the confidence we have in that difference. Given the KL divergence distance measure, we used standard hierarchal cluster analysis with average linkage. The clustering results from the analysis showed three major distinct cell clusters. The cluster analysis based on parameter values showed ~150-fold better cluster separation comparing to clustering based on calcium response directly (Fig EV1C–E). The identification of these clusters suggests the existence of three distinct cell states within the population. These cell

states are not dependent on cell cycles, cell size, and several other cellular morphology features (Fig EV3).

The use of kinetic parameters as signatures for cellular states enabled direct interpretation of mechanistic differences between the identified clusters. To examine how different parameters contributed to the clustering, we investigated the difference between the individual parameters among the three main clusters and how they related to the mechanism of calcium signaling. Figure 5A illustrates the degree of difference between clusters along different components of the pathway. We identified that the key parameters that separate these classes were related to the strength of positive and negative feedback between calcium and the IP3R channel, as well as the sensitivity of the IP3R channel to IP3 (Fig 5B). In our model, the IP3 channel is composed of three components: the IP3-activated subunit, the calcium-activated subunit, and the calcium-inactivated subunit. The significant parameters corresponded to the dissociation constants for these three subunits. Calcium has both activating and inhibiting effects on IP3R channels (Li & Rinzel, 1994). We found that the high magnitude of calcium responses in a cluster corresponded to stronger

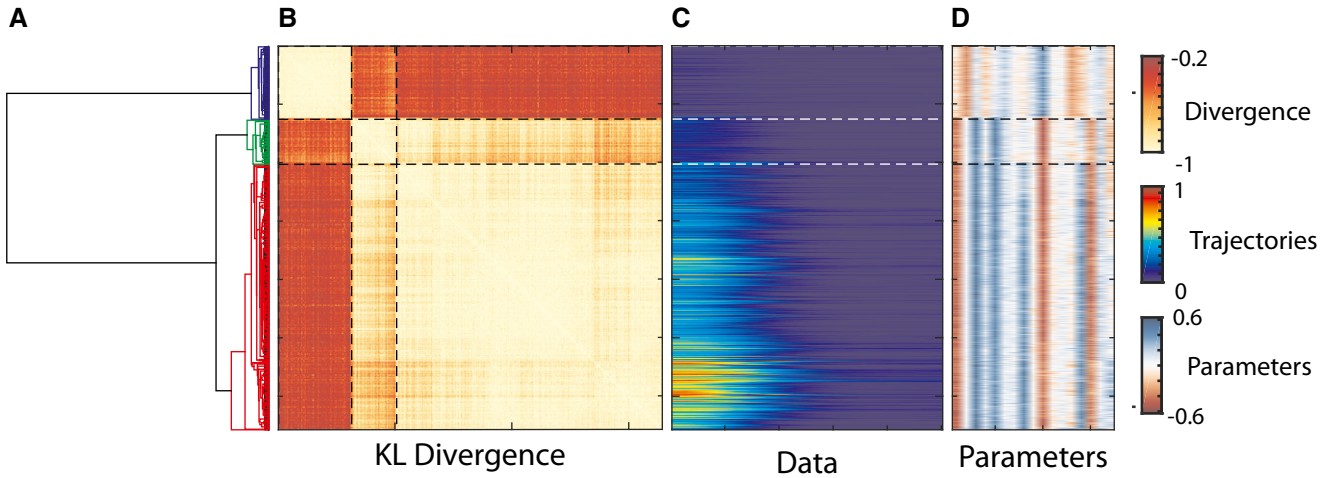

**Figure 4.  Clustering parameter distributions.**

A–D  The clustering results of parameter distributions based on the calculated Kullback–Leibler divergence measure. From the left to right are the (A) dendrogram of the clustering results, (B) distance matrix between parameter groups of single-cell data, (C) matrix of corresponding single-cell data, and (D) matrix of corresponding average parameter vectors for the single-cell data. Throughout the entire figure, each row represents the same cell.

Source data are available online for this figure.

activation and lower inhibition. Overall, the three clusters of parameter distributions were separated into a cluster with strong positive calcium feedback, a cluster with weak positive calcium feedback, and a third with strong negative calcium feedback (Fig 5C). As expected, each of the identified feedback structure had distinct temporal features associated with it (Fig EV4B). Our analysis makes a specific prediction that majority of cellular response variability is determined by IP3R activity. To test this prediction, we performed longitudinal experiment where cells were first exposed to extracellular ATP followed by increase in intracellular IP3. Intracellular increase in IP3 was achieved by 365 nm uncaging of preloaded caged IP3 (Materials and Methods). Our experiments show that ~70% of variability in magnitude of calcium response to ATP can be explained by response to intracellular IP3 (Fig 5D).

## Discussion

We employed an inference-based approach to gain insights into the structure of cellular state distributions. While cell-to-cell variability in cellular signaling responses is commonly observed, little is known about the statistical distributions of cellular responses. By fitting single-cell calcium responses to a mechanistic model using a Bayesian algorithm, we were able to gain insights into the structure of cell state distribution. By analyzing the distributions of the identified kinetic parameters, we found that cellular state distribution for calcium signaling was best approximated by a mixture model of three distinct clusters, represented by parameter distributions. Such characterization of cellular state space distribution is a fundamental step toward further work aimed at understanding the determinant of the observed signaling variability and its functional role.

Previous work proposed two possible hypotheses that can explain cell-to-cell extrinsic variability (Shibata & Fujimoto, 2005; Snijder *et al*, 2009; Ladbury & Arold, 2012; Rhee *et al*, 2014). The first hypothesis states that the variability is due to the accumulation of stochastic differences in gene expression, organelle structure, and other cellular processes that cause cells to drift randomly in the cell state space. According to this hypothesis, cellular variability is best described using an unstructured population distribution. The second hypothesis states that cells converge onto certain specific attractors in the cell state space instead of drifting randomly. Under this hypothesis, the basis for differences between cells depends on the different functional needs of the cell population instead of the stochastic fluctuations in gene expression. In other words, the second hypothesis states that cellular heterogeneity has a definite structure. Our approach of single-cell data model fitting showed that the kinetic parameters which define signaling are composed of a mixture of parameter distributions, supporting the second hypothesis. The different clusters of parameter distributions correspond to different fundamental signaling behaviors. The identification of these clusters raises interesting questions such as what is the physiological significance of the identified clusters? And what are the molecular mechanisms that enable the identified functional diversity?

In all likelihood, the structure of the cell state distribution is not universal but context dependent. In our work, we utilized the calcium response of cells to extracellular ATP as a model to investigate cell-to-cell variability. Extracellular ATP is a damage-associated molecular pattern (DAMP) that is released in response to wounds to activate neighboring cells (Handly *et al*, 2015). Response to ATP is one of the first steps in a very complex process of wound response and healing. It is plausible that during wound healing, there are different functional requirements for epithelial cells and therefore the observed variability is possibly a result of the need to generate functional diversity to enable multiple distinct roles (Sonnemann & Bement, 2011). Future work is needed to determine the degree of functional heterogeneity of epithelial cells during wound healing and whether any physiological functional diversity indeed corresponds to the identified clusters.

The functional diversity we identified can be mechanistically explained by changes in a few parameters within the calcium signaling pathway. IP3R channels on the ER play a key role in calcium signaling and have been shown to be responsible for the large diversity of calcium dynamic signaling responses to large array of ligands. Furthermore, IP3R channels are subjected to positive and negative feedback regulation by calcium (Lemon *et al*, 2003). In our model fitting results, the most notable variation in parameters among the three clusters of cell states is those in the parameters that govern the affinity of IP3R channels to both IP3 and the calcium ion. This key prediction was validated experimentally to show that indeed response to IP3 is the key difference between cells and by itself can explain 70% of the observed variability. Multiple mechanisms can cause such parameter differences including differences in expression of different subtypes of the IP3R (Wojcikiewicz, 1995) or in the structure of the ER that have an important role in the spatial distribution of IP3R channels (Meldolesi & Pozzan, 1998). What causes different cells to have different IP3R responses is still unclear. Like many cases of "symmetry breaking", it is possible that separation of cells into the three clusters of IP3R activity starts with stochastic events that get stabilized to create the emergence of three distinct populations. In addition to differences in IP3R regulation, a few other parameters showed differences between the three clusters such as receptor activity level (Fig EV4A). The "negative feedback" cluster had the highest level of receptor activity. However, in this cluster, due to the strong negative feedback, the low response was "robust", that is, insensitive to receptor levels. Therefore, the parameter levels of the receptor have minor effects on cellular response. This result exemplifies the need to analyze parameter sensitivity at the single-cell level because the sensitivity of receptor activity parameters varies in different cells based on their cluster identity.

We used an inferred kinetic parameter signature to define cell state and the subsequent analysis of these signature distributions indicated that the cellular population is composed of a mixture of three distinct cell states. Other signatures for cell state have been used in the past including single-cell genomics (Trapnell, 2015), single-cell cytokine secretion (Lu *et al*, 2015) and mass cytometry (Spitzer *et al*, 2015). The identification of distinct subpopulations is common across all these different definitions of cellular signature. Each one of these cellular signatures only captures some aspect of true cell state, and the recurrence of a complex population structure indicates that the high level of cellular variability is a result of multiple cell states that coexist in a population. In addition, our work and that of others suggest that these subpopulations are not static but dynamic states among which cells can potentially transition (Bendall *et al*, 2014; Marco *et al*, 2014; Durruthy-Durruthy & Heller, 2015; Setty *et al*, 2016). It will be interesting to apply a similar

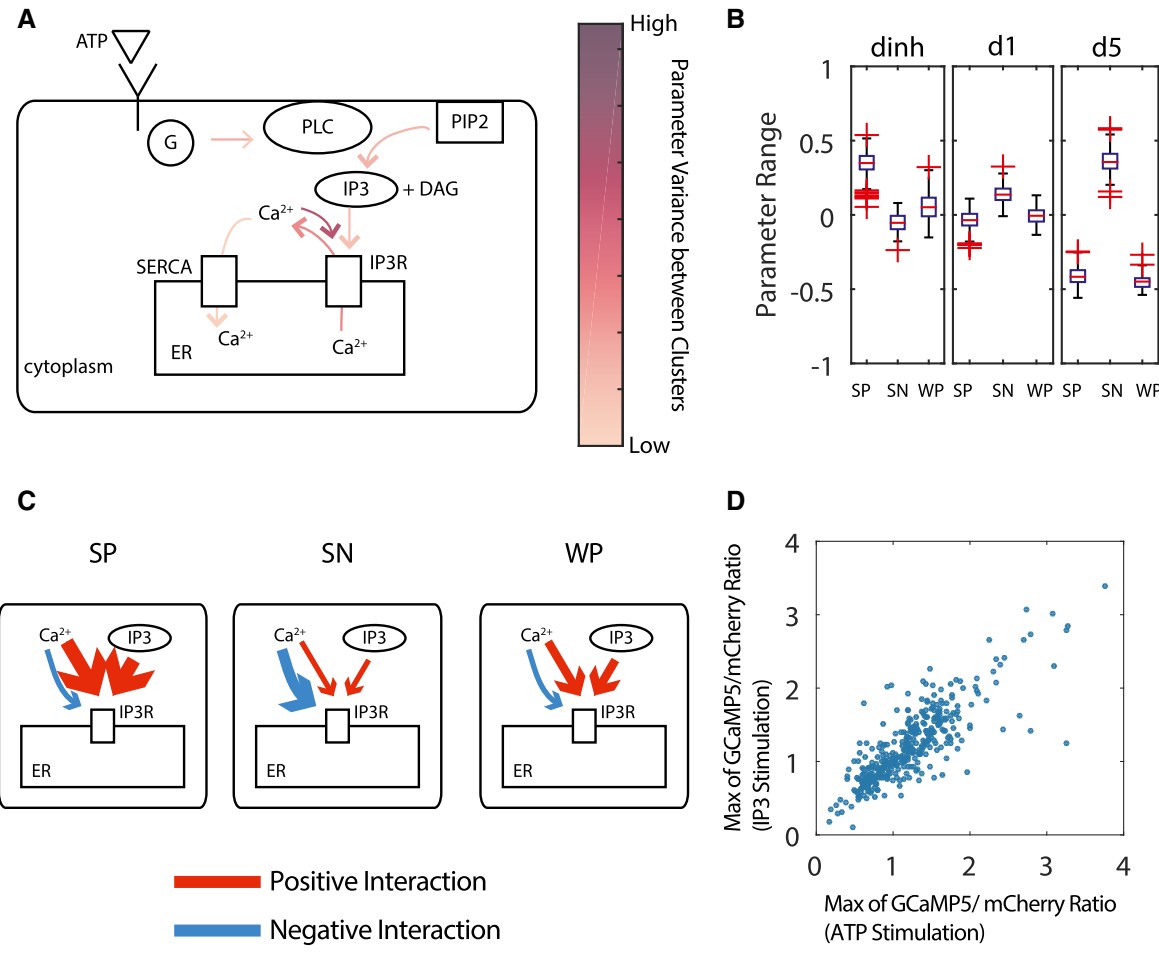

**Figure 5.  Mechanistic difference between clustered data.**

A   Calcium signaling schematic diagram showing the difference between parameters of the clusters as reflected in different components of the signaling pathway. The intensity of the arrows in the diagram corresponds to the degree of difference in the associated parameters between the three identified clusters.
B   A boxplot of the parameters of IP3R channels of the three major clusters corresponding to the equilibrium constant of the inactivating subunit of calcium channel (dinh) and equilibrium constant of activating subunit of IP3R (d1 and d5). SP, strong positive; SN, strong negative; WP, weak positive. The values of the parameter range values are the log of 10 with respect to the reference parameter values chosen as listed in Table EV1. The red lines of the boxplots represent the median of the distributions, and the ranges of the whiskers are defined to be 1.5 times the interquartile range (difference between 75th and 25th percentiles). Crosses are data outside of the interquartile range.
C   Illustrations of the differences in parameters as manifested in calcium signaling mechanics. The red color indicates positive feedback interaction, while the blue color indicates negative feedback interaction. The thickness of the arrows corresponds to the strength of those interactions.
D   The scatter plot between the maximum calcium responses to both the ATP stimulation and IP3 uncaging ($R^2$ = 0.68).

trajectory analysis to other kinetic parameter signature datasets to track how cells respond to ligand changes over time and to what degree these changes are reflected by transitions between the identi-fied distinct cell states.

   Prior to this work, the predominant method for fitting models to signaling pathways has been to fit only the population average and come up with one best fit parameter vector. The prevailing assump-tion of this traditional method is that the population average data are a good representation of the overall behavior of the cell popula-tion, and therefore, the best fit parameter vector is also the best representation of the cell population at the mechanistic level. While experimental work has consistently demonstrated the limitations of population averaging, mathematical modeling has lagged behind by treating of cells in the population equally and modeling only the average behavior. Difficulties in parameter inference due to model non-identifiability (Raue *et al*, 2011) or parameter "sloppiness" (Gutenkunst *et al*, 2007) further limit the ability to perform such parameter fitting at the single-cell level. Our work demonstrates that fitting parameters at the single-cell level are useful because it prevents information loss due to population averaging and enables the discovery of distinct subpopulations. The improved performance is despite the fact that, at least in some of the parameters, the confi-dence interval of the identified parameter values is quite large and therefore the fit could be considered "sloppy". Because the presented framework is independent of the pathway in considera-tion, it can be applied to other signaling pathways which also show considerable cell-to-cell variability in many other signaling networks. It will be interesting to see whether the structure

distribution and existence of multiple distinct cell states are universal and occur in other signaling networks.

# Materials and Methods

### Cell plating

MCF10A cells were cultured in F12/DMEM media supplemented with horse bovine serum, insulin, cholera toxin, and hydrocortisol according to (Debnath *et al*, 2003). Before the ligand perturbation experiment, cells were cultured overnight in a 96-well plate in assay media as described in Debnath *et al* (2003). On the day of the experiment, cells were incubated with a solution of 4 μM Fluo-4 (a calcium indicator) and 1 μM Hoechst (a nucleic acid dye) for 20 min. After these 20 min, we exchanged the media with extracellular Hepes buffer supplemented with probenecid and glucose.

### Preparation of calibration buffer

To calibrate the concentration of cytosolic calcium from the raw intensity, we prepared calibration buffer solutions according to an established protocol (Bao *et al*, 2010). The minimum calibration buffer was comprised of 50 μM ionomycin, 5 μM thapsigargin, and 12.5 mM EGTA; the maximum calibration buffer was comprised of 50 μM ionomycin, 5 μM thapsigargin, and 36 mM $CaCl_2$. Both buffers were in EC buffer supplemented with glucose and probenecid.

### Measurements of single-cell calcium response to an increase in extracellular ATP perturbation

Image acquisition of the 96-well plate was conducted on a Nikon Ti microscope using a 10× objective (Selimkhanov *et al*, 2014). The microscope was automated using micro-manager (www.micro-manager.org) through its MATLAB scripting interface. After acquiring baseline calcium levels for 5 min, 10 μM ATP was added and acquisition continued for another 15 min. To allow single-cell calibration of the fluorescent response, the extracellular buffer was washed three times and replaced with minimum calibration buffer. Acquisition continued until equilibration of calcium levels followed by additional washes and the addition of maximum calcium buffer. The images were segmented to locate the Hoechst signals as the positions of the nuclei. The calcium signals were mapped to the Hoechst images taken at different time points to establish single-cell trajectories. The procedure is identical to that described in Selimkhanov *et al* (2014). Concentration of calcium was calibrated following Kao *et al* (1989). Briefly, for each single-cell, the $F_{min}$ and $F_{max}$ were estimated following treatment with the minimal and maximal calibration buffers, respectively. Calcium levels were calculated as: $[Ca^{2+}]_{free} = K_d \frac{(F-F_{min})}{(F_{max}-F)}$ where $F$ is the intensity measurement and $K_d$ is the dissociation constant between Fluo-4 and calcium.

### Longitudinal dual-stimulation experiment

Dual-stimulation experiments were performed by first stimulated with ATP and then with intracellular IP3 by uncaging a previously loaded caged IP3 in MCF10A cells that stably express genetically encoded calcium activity reporter GCaMP5 (Akerboom *et al*, 2012) fused to mCherry (Su *et al*, 2013). The mCherry was used to normalize for expression level by dividing the GCaMP signal for each cell by the average mCherry signal over the same segmentation mask. Prior to experiments, cells were first incubated in 1:2,000 dilution of 10 mg/ml Hoechst and then incubated in 1.5 μM membrane permeable caged IP3 for 1 h (Enzo Life Sciences, Yan *et al*, 2015). During the experiment, the cells were first stimulated with 10 μM ATP and imaged for the subsequent 15 min. After the 15-min period, cells were allowed 35 min to recover from the ATP stimulated while still imaged to enable cell tracking. After the recovery period, cells were exposed to 365-nm LED light (Thorlabs) for 5 s to uncage IP3. The subsequent calcium response from the cells was imaged for 5 min.

### Model of calcium signaling

Our calcium model (Fig 2) is based on a compilation of models from Lemon *et al* (2003) and Li and Rinzel (1994) with few simplifications to improve parameter estimation through the increase of model identifiability. We simplified the receptor dynamics to an ATP-dependent step function followed by first-order decay. This was implemented by multiplying the activation of PLC by receptor and ligand concentration L and an exponential term which accounts for the desensitization of the surface receptor. The equation that describes the rate of change of IP3 is a Hill function which describes the production rate of IP3 in terms of PLC. The degradation rate of IP3 is a linear rate term proportional to IP3 concentration. The description of calcium dynamics was modified from Lemon *et al* (2003) to simplify and have only one term for the calcium buffer. The equation for the fraction of activated IP3 channels was taken from the simplified model of Li and Rinzel (1994). The model equations are presented in Fig 2, and parameters are in Table EV1.

### Parameter estimation: pre-processing of calcium response

Cellular calcium responses contained stochastic elements, and the experimental noise was not captured by the mathematical model described above. To improve parameter estimation, we pre-processed the single-cell calcium response data to remove aspects of the responses that were not captured by our model. We used a low-pass filtering scheme. Because the initial calcium response to ATP was very sharp, we truncated the baseline measurement to include only frames following ATP addition and utilized a "reflective boundary condition", that is, we concatenated to the signal a mirror image of itself prior to filtering. The time points before the addition of ATP are not included in the model fitting and are used only to calculate the basal level point, which was designated as the first time point of each trajectory. As a result of this pre-processing, we removed all the high-frequency elements in calcium response that were not captured by our model without dampening the initial sharp response.

### Parameter estimation: calculating goodness of fit

The goodness of fit between the simulated data and real data was determined by performing a sum square of error on the comparison

between a simulation and the smoothed experimental data. We then compared the first derivative of the simulation and the first derivative of the smoothed experimental data. We found that including information on the first derivative improved the fitting as it increased the emphasis on the shape of the curve compared with traditional sum square error. We used the difference between the raw (unprocessed) and the processed data as an acceptance threshold to be considered a "good fit". In other words, if the simulated data agree with raw data with the same or better goodness of fit than simple smoothing, then the parameters corresponding to the simulated data will be accepted as a good fit.

### Parameter estimation: rejection sampling

We utilized a SMC-ABC parameter estimation following Liepe *et al* (2014). At the core of the SMC-ABC algorithm is a rejection sampling estimator that estimates posterior distribution based on a prior ($\pi(\Theta)$), a goodness-of-fit score (**score(D',D)**), and an acceptance threshold ($\varepsilon$). A parameter vector is defined by the collection of parameters that are needed to fully specify the model shown in Fig 2. To sample an *N* parameter vectors from the posterior distribution, the algorithm randomly samples parameters from the prior and simulates them and scores their agreement with the data; it then accepts only the parameters that fit to the data within the threshold score. The rejection sampling pseudocode is shown in Scheme 1.

### Parameter estimation: approximate Bayesian computation sequential Monte Carlo

The challenge with the rejection sampling described above is that if the prior is very different from the posterior, the efficiency of sampling can be too low to be practical. The ABC-SMC algorithm circumvents this problem by repeating the rejection sampling iteratively where in each iteration, the prior distribution and the acceptance threshold are updated. The algorithm performs the first iteration of rejection sampling with the prior distribution $\pi_0(\Theta)$ provided by the user. In the actual implementation, $\pi_0(\Theta)$ is a log10 uniform distribution in the range $[-1\ 1]$ of the parameter value fold difference from the reference values of the parameters listed in Table EV1. The first iteration terminates when the sampling procedure collects a parameter vector population of size $N_{iteration}$ which meets the preliminary threshold requirement $\varepsilon_{prelim}$. The threshold score for each of the intermediate iterations of rejection sampling is determined relatively. For a parameter vector to be accepted, it has to produce simulated data that score better than the bottom 90% fraction of the population collected in the previous iteration. The entire algorithm terminates when it reaches the desired number of samples, all with goodness-of-fit scores lower than the threshold $\varepsilon_{final}$. The threshold $\varepsilon_{final}$ is determined by scoring the raw data against the processed data. Therefore, the threshold $\varepsilon_{final}$ is determined for each single cell based on the data quality for that cell.

---

**Input:**

$\pi(\Theta)$ = The prior probability distribution from which the parameter vector $\Theta$ is drawn

**N** = The size of the parameter vectors needed for the posterior in each iteration of rejection sampling

$\varepsilon$ = The goodness of fit score needed for the parameter vector to be accepted into the posterior

**D** = The data to be fitted with the model

**score**(D',D) = function to score the goodness of fit of D' against D

**Output:**

**Q** = the population of accepted parameter vectors as an approximation of $\pi(\Theta|\textbf{score}(D_{sim},D) < \varepsilon)$, the posterior distribution of the parameters

**Begin**
Initialize **Q** to be the empty set
**While |Q| < N**
 Sample parameter vector $\Theta'$ from $\pi(\Theta)$
 Carry out simulation with $\Theta'$ to produce simulated data $D_{sim}$
 Score $D_{sim}$ against **D**
 **If score($D_{sim}$,D) < $\varepsilon$**
 Add $\Theta'$ to **Q**
 **EndIf**
**EndWhile**
**End**

**Scheme 1.  Rejection sampling.**

**Parameter estimation: convergence test**

A key challenge in parameter fitting is the possibility that the fit will identify a "local minimum" that does not represent the full distribution of possible parameters. To test that this is not the case, we utilized a strict test for convergence. For each cell for which we attempted to estimate its parameters, the entire ABC-SMC algorithm described above was performed twice in two completely independent runs (run #1 and run #2) to produce two samples of parameter vectors. To determine whether these two samples were statistically indistinguishable, we performed the following test: For each of the single-cell data, we combined the parameters from both runs and for each parameter identified the run identity (e.g., #1 or #2) of the nearest neighbor to calculate the probability that a nearest neighbor has the same run identity as the tested parameter. We then compared this measured distribution with a "null" model where there was equal (0.5) probability that a sample nearest neighbor was from the same run. If the chi-square statistic failed to reject the null hypothesis with significance level 0.05, we determine that the two populations are convergent.

**Clustering of the parameter ensemble**

To cluster cells by calcium response, we first identified a suitable distance measure. Our estimation for each cell state was a sample from the posterior parameter distribution given the model and the data from that cell. Therefore, we have not only information on the degree of overall difference between two parameter sets, but also degree of variability of parameters within this set. We designed a distance measure that takes this parameter uncertainty into account. Specifically, for every two cells $i$ and $j$, we first construct a distribution that quantify the probability that a specific parameter vector for cell $i$ ($j$) has a nearest neighbor parameter vector that belong to the same cell $i$ ($j$). We then calculated the Kullback–Leibler (KL) divergence of the measured probability distribution from a null model where identities $i$ and $j$ for all parameter samples are randomly permuted. The permutation reflects a null model where there is no difference between cell $i$ and $j$ in their parameter distributions The KL divergence from a random model was used as the distance measure for cluster analysis. The pairwise KL divergence was then calculated for all the ensembles (Fig 3). We then used hierarchical clustering with average linkage to cluster the ensembles based on the KL divergence measure. The hierarchical tree was subsequently thresholded to generate clusters based on Calinski–Harabasz optimal criterion that used medoids of individual parameter ensembles associated with single cells as the center of those ensembles. In the same way, the medoids of clusters of parameter ensembles were used to represent the cluster centers (Caliński & Harabasz, 1974).

**Data availability**

The mathematical model from this publication has been submitted to BioModels Database and assigned the identifier MODEL1611150001. The same SBML file is also attached as Computer Code EV1 along with this paper.

**Expanded View** for this article is available online.

## Acknowledgements

We thank Noa Pinter-Wollman for critical reading of the manuscript. The work was supported by GM111404 and EY024960 from NIH.

## Author contributions

JY and RW designed the study and performed analysis. JY performed the experiments. AP made key reagent. JY and RW wrote the manuscript.

## Conflict of interest

The authors declare that they have no conflict of interest.

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
