## [Review Process File · Molecular Systems Biology]

Distinct Cellular States Determine Calcium Signaling Response

Jason Yao, Anna Pilko and Roy Wollman

Corresponding author: Roy Wollman, UCLA

Review timeline:

Submission date:	15 June 2016
Editorial Decision:	27 July 2016
Revision received:	21 October 2016
Editorial Decision:	11 November 2016
Revision received:	16 November 2016
Accepted:	18 November 2016

Editor: Maria Polychronidou

Transaction Report:

1st Editorial Decision

27 July 2016

Thank you again for submitting your work to Molecular Systems Biology. We have now heard back from two of the three referees who agreed to evaluate your study. Unfortunately, after several reminders we have not managed to obtain a report from reviewer #3. Since the recommendations of the other two referees are similar, I prefer to make a decision now rather than further delaying the process. As you will see below, the reviewers acknowledge that the study is interesting and that the presented methodology is likely to be useful for the field. However, they raise a number of concerns, which should be carefully addressed in a major revision of the manuscript.

Without repeating all the points listed below, some of the more fundamental issues are the following:

- The conclusion regarding the existence of distinct states needs to be strengthened by additional analyses.
- Providing further experimental evidence related to the mechanisms giving rise to the three cells states would significantly enhance the impact of the study.

REFeree REPORTS

Reviewer #1:

Review of 'Distinct Cellular States Determine Calcium Signaling Response' by Jason Yao and Roy

Wollman.

In this manuscript, the authors perform timelapse imaging of calcium response to ATP in single MCF10A cells to obtain single-cell timetraces. They then perform mathematical modelling combined with a Bayesian Inference-based fitting approach to identify and fit sets of differential equation systems to each single cell, to determine parameter values for each single cell that result in models that describe pre-processed single-cell traces best. Finally, they calculate the divergence between the probability distributions of these parameters of two cells in a pairwise manner across the whole dataset using a method from information theory (Kullback Leibler divergence), and use this, in combination with the actual distributions, to cluster single cells, revealing three major cell clusters. These three clusters are then interpreted as providing possible mechanistic insight into the nature of these three clusters, namely the existence of strong positive feedback of calcium on the IP3 receptor, weak positive feedback, and negative feedback.

Generally, I found the manuscript easy to follow, it is well written, explains the ideas and methods well, and communicates the results clearly. The overall impression is that this is a simple and elegant approach, which appears to work well. The findings provide support for one of the two main views on how cell-to-cell variability arises in isogenic mammalian cell populations exposed to the same conditions, namely from the existence of distinct cellular states (or state distributions) which are determined by differences in cellular physiology and not caused by random or stochastic drifts. As such, the work is an important contribution and adds to the increasing body of literature that favors this more deterministic view. It also provides a potentially very useful and relatively straightforward method to a large number of researchers that are analyzing single-cell time traces of dynamic cellular responses, which may yield surprising insights, notably in the cell signaling field where the prevailing view still is that most variability is of a stochastic nature.

While the above is all positive, I do have some major criticisms, which I summarize below:

1. The overall impression of the paper is that it is a bit 'thin'. There is no experimental validation of the three states. The paper would significantly strengthen if such work would be included. Could the authors come up with measurements of the strength of positive feedback in single cells? Could they explore the use of IP3R inhibitors or agonists to dissect this? I am not an expert on calcium signaling, but it appears to me that many experimental tools are available for this system to obtain experimental evidence for the proposed mechanisms that underlie the three major cell states. With such data included, the paper would be much stronger.
2. As the authors state, they pre-process the raw time-traces before they attempt model fits, and they also somewhere mention low-pass filtering of the data. Clearly, this removes some of the variability in the data. An alternative approach is to ask how much of the total variability in the data (without any pre-processing) can be explained by the models. And after removing that, is the amount of unexplained variability approaching the Poisson limit, plus some experimental noise? Can the amount of experimental noise in their setup be estimated?
3. Finally, while the distinct cellular states support the more deterministic view, the authors provide no insight in what is the source of these different states? In theory these could still emerge from stochastic events occurring in growing cell populations which then become stabilized. The authors suggest from Figure 1f-g and extended view Figure 3 that it is not correlated with the spatial positioning of cells or with the amount of DNA in cells (used as a proxy for position in the cell cycle). This analysis is however underwhelming. Merely glancing at an image of a couple of hundred cells does not provide any quantitative insight about whether there are correlations with local cell density, cell size, or cell shape. Furthermore, Hoechst staining can be tricky to interpret (depending on amount and time of incubation). Do the authors see two nice peaks of G1 and G2 cells in Hoechst intensity distributions? Providing more insight into what could be the sources that determine the 3 major cellular states would also be a major improvement of the manuscript.

Reviewer #2:

Yao & Wollman address an interesting and timely question: to what degree the experimentally observed single-cell variability results from purely stochastic biochemical events and/or whether discrete states exist in a population. They address it by measuring single-cell calcium responses to ATP and by further fitting the responses to a previously published model.

Overall, the methodology presented in the paper is a valuable addition to the field, however, the conclusions about the existence of discrete states in the parameter space are not convincing given the current presentation. Thus, before recommending the paper for publications the authors should address the following.

Major issues:

1. The clustering of distance measure based on KL divergence shown in Fig.4 implies 3 distinctive groups in the parameter space. At the same time, authors conclude that hierarchical clustering of measured responses shown in EV Fig.1A doesn't yield good grouping. One issue I have with this claim is that just by looking at the heat-map/waterfall plot EV Fig.1A I can discern ~6 clusters, based mainly on the time to reach the final steady-state value.

Regardless of authors' claim about the lack of "natural grouping" of experimental time courses, it is unclear to me why they insist there wouldn't be a (decent) agreement between explicit clustering of experimental time courses and clustering of model parameters fitted to those very same time courses. After all, the model with parameter sets from different clusters shown in Fig.4 should generate time courses different enough to classify/assign them to a couple of distinct clusters. If that wasn't the case, the robustness of fitting would be in question.

In other words, distinct states in the parameter space that authors attribute to different feedback mechanisms should relate to distinct dynamic responses. This is partly addressed in Fig.4C but the "green" cluster contains all kinds of dynamic responses, from weak to strong activation. How does it relate to explicit clustering of measured responses (based on entire time courses or their features)?

Additionally to addressing the above concerns, I ask the authors to:

(a) provide a confusion matrix of clustering of raw (or smoothed) measured responses against classification of fitted parameters. How similar percentage wise are those two classifications?

(b) expand Fig. 4 with a panel including time course features (according to Fig. 1D). Are those features distinct between 3 clusters?

(c) add labels designating feedback mechanism to the 3 clusters.

2. Related to the above is the question about response dynamics within three clusters that were associated with different feedback strengths and types. For example, it is known that due to the negative feedback the dynamic responses are transient, while strong positive feedback can induce sustained responses. Thus, are responses belonging to the "Neg. Feedback" cluster more transient (decay faster) than those in strong and weak PF?

If no such differences exist in time courses that belong to any of 3 clusters, then the mechanistic explanation of these clusters is questionable.

3. The following sentence is unclear:

"For every two parameter vector populations corresponding to each of two cells, the distance measure was defined as the KL divergence from a null model where both samples were obtained from identical probability distributions."

Please rephrase/expand the definition of the distance. In particular, please comment on the validity of KLdiv usage in this context, given that KLdiv isn't a true metric, i.e. (usually) $KLdiv(a || b) \neq KLdiv(b || a)$.

Also, the explanation in M&M - "Clustering of the Parameter Ensemble" is not clear and in its current form one can easily get lost in terminology. The entire section would greatly benefit from a

scheme of the work-flow, or an itemised step-by-step explanation.

4. Fig. 1E

The authors have excluded the influence of spatial localisation and cell cycle. However, part of the observed variability could still result from pre-existing cellular states. For example high basal levels (F_0) could imply high peak value (F_m). To further exclude those effects authors should provide scatter plots for pair-wise comparison of 6 main time course features. Are they correlated?

Minor issues:

1. Fig. 1A.

How many cells have been measured in a single experiment?

What is the colour coding scale associated with this panel?

2. Fig. 1E.

The range of Y-axis scale makes it difficult to discern the boxes. Consider removing two outliers in the last two rows, or skipping the last two columns with ratios altogether.

Please add the equation used to calculate the Z value.

3. EV Fig.1B

The visualisation doesn't seem to be appropriate. 6 features along the x-axis transition smoothly; it's hard to recognise that the axis is discrete.

4. EV Fig.3

Current box plot visualisation makes it difficult to see whether G1, S, G2/M are indeed equally populated across three clusters. Thus, I suggest to change this plot or to add a panel with a more familiar cell cycle plot, i.e. line plot of the probability density of Hoechst intensity for 3 different feedback types/strengths.

5. Fig. 3D

What kind of normalisation was used? Area, count, width? Please describe in the caption. The sentence from the caption: "The width of each violin shows the probability density of the underlying distribution." is unclear.

6. EV Fig.4

What does it mean that whiskers do not extend from the box? E.g. " η_1 (ER)" for WP.

What kind of box-plot is that? What do whiskers signify?

7. Sentence, page 9, par. 1:

"To test if the nearest neighbours probability that we calculated shows a statistically significant different..." -> difference

8. Table S1 was missing from my copy of the manuscript.

9. Please explicitly define what "parameter vector" means.

10. Since the model was simplified compared to the original, please deposit the model into BioModels database.

We are excited to re-submit a revised manuscript for your consideration. The paper was revised based on the reviewer comments and now includes more detailed analysis and new experimental data. Below is point by point response to reviewer comments. For clarity we show reviewer comments in serif fonts italics and our response in san serif below.

Editorial comments

- The conclusion regarding the existence of distinct states needs to be strengthened by additional analyses.

We enhanced the basic cluster analysis presented in this work in the following way. In the original analysis the final cluster “cutoff” was determined based on what we subjectively considered to be “natural clusters”. In the revised submission we now use formal criteria for cluster number identification based on modified Calinski-Harabasz criteria. The use of formal criteria improves the analysis in two important ways: 1. it removes some of the subjectivity in the procedure. 2. It allows formal comparison between clustering based on parameter inference to clustering based on calcium responses directly. These comparisons show that while there is some similarity between the groupings based on different clustering approaches, the quality of clustering based on parameter inference is much higher than any other approach.

- Providing further experimental evidence related to the mechanisms giving rise to the three cell states would significantly enhance the impact of the study.

Our analysis identified IP3R activity as a key factor that distinguishes the three cell states and determines calcium response. Therefore this analysis predicts that response IP3 should explain the majority of cellular response variability. This is a non-trivial prediction as it was not clear a-priori that this is where majority of the variability is and not for example in receptor level. We experimentally tested this prediction with a new experiment where we stimulate cells with ATP, allow them to recover for 30 min and then uncage using light a caged IP3. We show that the amplitude of calcium response to IP3 explains ~70% of the variance in amplitude of calcium response to ATP. The new data (shown in Figure 5d) provides a validation of our key finding in way that is independent on any parameter fitting.

Reviewer #1:

Generally, I found the manuscript easy to follow, it is well written, explains the ideas and methods well, and communicates the results clearly. The overall impression is that this is a simple and elegant approach, which appears to work well. The findings provide support for one of the two main views on how cell-to-cell variability arises in isogenic mammalian cell populations exposed to the same conditions, namely from the existence of distinct cellular states (or state distributions) which are determined by differences in cellular physiology and not caused by random or stochastic drifts. As such, the work is an important contribution and adds to the increasing body of literature that favors this more deterministic view. It also provides a potentially very useful and relatively straightforward method to a large number of researchers that are analyzing single-cell time traces of dynamic cellular responses, which may yield surprising insights, notably in the cell signaling field where the prevailing view still is that most variability is of a stochastic nature.

We thank the reviewer for his/her appreciation for the conceptual and methodological contribution of this work.

While the above is all positive, I do have some major criticisms, which I summarize below:

1. The overall impression of the paper is that it is a bit 'thin'. There is no experimental validation of the three states. The paper would significantly strengthen if such work would be included. Could the authors come up with measurements of the strength of positive feedback in single cells? Could they explore the use of IP3R inhibitors or agonists to dissect this? I am not an expert on calcium signaling, but it appears to me that many experimental tools are available for this system to obtain experimental evidence for the proposed mechanisms that underlie the three major cell states. With such data included, the paper would be much stronger.

We now provide experimental validation of key prediction that stems from our analysis. Our analysis identified IP3R activity as key parameter that we predict should explain the majority of calcium response variability. We now include data from new experiment to directly test this prediction (new panel Figure 5d). In this new experiment we performed longitudinal analysis where we measured for the each cells the response to extracellular ATP and intracellular increase in IP3. We show that ~70% of the response variability in calcium response magnitude is explained by the magnitude of the response to IP3. This new data validates the key prediction of the computational analysis regarding the key mechanisms that explain IP3 response as shown in Figure 5d.

2. As the authors state, they pre-process the raw time-traces before they attempt model fits, and they also somewhere mention low-pass filtering of the data. Clearly, this removes some of the variability in the data. An alternative approach is to ask how much of the total variability in the data (without any pre-processing) can be explained by the models.

We now include new expanded view panel (Expanded View Figure 2e) that show that our initial pre-processing removes only a small fraction of the observed variability.

And after removing that, is the amount of unexplained variability approaching the Poisson limit, plus some experimental noise? Can the amount of experimental noise in their setup be estimated?

The observed variability between cells was significantly higher than experimental variability. We now included new expanded view panel (EV Figure 3a) that quantify and compared the experimental variability (that our pre-processing is aimed to remove) and the observed cell to cell variability. As the process is dominated by cellular heterogeneity is kinetic parameters, the observed variability is much higher than would be expected by simple Poisson process with fixed parameters.

3. Finally, while the distinct cellular states support the more deterministic view, the authors provide no insight in what is the source of these different states? In theory these could still emerge from stochastic events occurring in growing cell populations which then become stabilized.

We agree with the reviewer and now added the following text in the discussion paragraph that discusses the three distinct IP3R mechanism:

“What causes different cells to have different IP3R responses is still unclear. Like many cases of “symmetry breaking” it is possible that separation of cells into the three clusters of IP3R activity start with stochastic events that get stabilized to create the emergence of three distinct populations”

The authors suggest from Figure 1f-g and extended view Figure 3 that it is not correlated with the spatial positioning of cells or with the amount of DNA in cells (used as a proxy for position in the cell cycle). This analysis is however underwhelming. Merely glancing at an image of a couple of

hundred cells does not provide any quantitative insight about whether there are correlations with local cell density, cell size, or cell shape. Furthermore, Hoechst staining can be tricky to interpret (depending on amount and time of incubation). Do the authors see two nice peaks of G1 and G2 cells in Hoechst intensity distributions? Providing more insight into what could be the sources that determine the 3 major cellular states would also be a major improvement of the manuscript.

We expanded the analysis and analyzed 12 different cellular features that include cell size, morphology, and local neighborhood. The analysis is shown in expanded figure 3b. We did not identify any simple factor that can explain the separation into three clusters. Any “negative result” in this type of analysis is inconclusive as it is always possible that we did not include the “right” feature. What causes cells to separate into the observed clusters is a key open question. And while a fascinating one, we consider it to be outside the scope of the current work.

Reviewer #2:

Yao & Wollman address an interesting and timely question: to what degree the experimentally observed single-cell variability results from purely stochastic biochemical events and/or whether discrete states exist in a population. They address it by measuring single-cell calcium responses to ATP and by further fitting the responses to a previously published model.

Overall, the methodology presented in the paper is a valuable addition to the field, however, the conclusions about the existence of discrete states in the parameter space are not convincing given the current presentation. Thus, before recommending the paper for publications the authors should address the following.

We thank the reviewer for his/her constructive comments.

Major issues:

1. The clustering of distance measure based on KL divergence shown in Fig.4 implies 3 distinctive groups in the parameter space. At the same time, authors conclude that hierarchical clustering of measured responses shown in EV Fig.1A doesn't yield good grouping. One issue I have with this claim is that just by looking at the heat-map/waterfall plot EV Fig.1A I can discern ~6 clusters, based mainly on the time to reach the final steady-state value. Regardless of authors' claim about the lack of "natural grouping" of experimental time courses, it is unclear to me why they insist there wouldn't be a (decent) agreement between explicit clustering of experimental time courses and clustering of model parameters fitted to those very same time courses. After all, the model with parameter sets from different clusters shown in Fig.4 should generate time courses different enough to classify/assign them to a couple of distinct clusters. If that wasn't the case, the robustness of fitting would be in question. In other words, distinct states in the parameter space that authors attribute to different feedback mechanisms should relate to distinct dynamic responses. This is partly addressed in Fig.4C but the "green" cluster contains all kinds of dynamic responses, from weak to strong activation. How does it relate to explicit clustering of measured responses (based on entire time courses or their features)?

We completely agree with the reviewer comments that:

1. That the use of “natural grouping” is subjective.
2. It is possible to find clusters in the hierarchical clusters based on calcium response alone.
3. That there should be some agreement between clustering based on calcium response and parameters that are fitted to calcium response.

To enhance our analysis we now address these three points specifically. We replaced the subjective last step that determines the cutoff within the hierarchical tree with an objective criterion that is

based on modified Calinski-Harabasz criteria. We apply this criterion for the clustering based on the inferred parameters and the clustering based on calcium response directly. We then compared the quality of the identified clusters based on two approaches based on a measure that looks at the ratio of variance between clusters divided by variance within cluster. We now show (Expanded Figure 1 c,d) that the clustering based on parameter is ~150 times better than clusters based on calcium response directly. Therefore, while it is possible to identify clusters in any hierarchical tree, the parameter inference we performed substantially increase the quality of the identified clusters.

Additionally to addressing the above concerns, I ask the authors to:

(a) provide a confusion matrix of clustering of raw (or smoothed) measured responses against classification of fitted parameters. How similar percentage wise are those two classifications?

The confusion matrix is now shown in Expanded View Figure 1e.

(b) expand Fig. 4 with a panel including time course features (according to Fig. 1D). Are those features distinct between 3 clusters?

The requested data is now shown in Expanded View Figure 4b. As pointed out by the reviewer, given that the three clusters have different underlying mechanism, and given that different mechanism should result in different time course features, one would expect to see that the time course features are distinct between three different clusters. This is indeed what we see as shown in Expanded View Figure 4b

(c) add labels designating feedback mechanism to the 3 clusters.

Labels that show how temporal features are distributed by cluster is now shown in Expanded View Figure 4b

2. Related to the above is the question about response dynamics within three clusters that were associated with different feedback strengths and types. For example, it is known that due to the negative feedback the dynamic responses are transient, while strong positive feedback can induce sustained responses. Thus, are responses belonging to the "Neg. Feedback" cluster more transient (decay faster) than those in strong and weak PF? If no such differences exist in time courses that belong to any of 3 clusters, then the mechanistic explanation of these clusters is questionable.

We clarify in the main text that we would expect to see, and indeed are seeing, differences in response dynamics between different clusters.

"As expected, each of the identified feedback structure had distinct temporal features associated with it (Expanded View Figure 4b)"

3. The following sentence is unclear:

"For every two parameter vector populations corresponding to each of two cells, the distance measure was defined as the KL divergence from a null model where both samples were obtained from identical probability distributions."

The sentence was rewritten to be:

"For every two cells we constructed a probability distribution of the likelihood of nearest neighbor parameter vector to be from the same cells. The parameter vector populations corresponding to each of two cells, the distance measure was defined as the KL divergence from the constructed distribution from a null model where the parameter vector sets for the two cells are both samples were fully mixed obtained from identical probability distributions."

Please rephrase/expand the definition of the distance. In particular, please comment on the validity of KLdiv usage in this context, given that KLdiv isn't a true metric, i.e. (usually) $KLdiv(a || b) \neq KLdiv(b || a)$.

As explained above, the KL distance was not between the two cells multivariate parameter distribution but between a probability distribution that capture the level of “mixing” between cells and a null model where the parameter distributions are fully mixed. This was done for two reasons: 1. as noted by reviewer, KL it not a true metric as it is not symmetric. 2. The dimensionality of the multivariate distribution was too high for direct comparison. The constructed statistics does capture the essence as it take into account the spread of parameters within cell as well as the separation between cells.

Also, the explanation in M&M - "Clustering of the Parameter Ensemble" in not clear and in its current form one can easily get lost in terminology. The entire section would greatly benefit from a scheme of the work-flow, or an itemised step-by-step explanation.

The section was completely revised for clarity.

4. Fig. 1E

The authors have excluded the influence of spatial localisation and cell cycle. However, part of the observed variability could still result from pre-existing cellular states. For example high basal levels (F_0) could imply high peak value (F_m). To further exclude those effects authors should provide scatter plots for pair-wise comparison of 6 main time course features. Are they correlated?

To clarify, we do not claim anywhere that calcium response dynamic is the same between the different clusters. This will be impossible as the clusters are based on kinetic parameters that are inferred from calcium response dynamics. Our key point, is that the parameters provide a better view of these differences that both shows higher degree of separation and is easier and more intuitive to analyze. We include new quantification that shows this point (Expanded View Figure 1b).

Minor issues:

1. Fig. 1A.

How many cells have been measured in a single experiment?

Added to figure legend.

What is the colour coding scale associated with this panel?

Colorbar included.

2. Fig. 1E.

The range of Y-axis scale makes it difficult to discern the boxes. Consider removing two outliers in the last two rows, or skipping the last two columns with ratios altogether.

Corrected.

Please add the equation used to calculate the Z value.

3. EV Fig. 1B

The visualisation doesn't seem to be appropriate. 6 features along the x-axis transition smoothly; it's hard to recognise that the axis is discrete.

Additional labels were added to clarify that the axis is discrete.

4. EV Fig.3

Current box plot visualisation makes it difficult to see whether G1, S, G2/M are indeed equally populated across three clusters. Thus, I suggest to change this plot or to add a panel with a more familiar cell cycle plot, i.e. line plot of the probability density of Hoechst intensity for 3 different feedback types/strengths.

We now include the Hoescht analysis together with additional 11 other features in a different panel.

5. Fig. 3D

What kind of normalisation was used? Area, count, width? Please describe in the caption. The sentence from the caption: "The width of each violin shows the probability density of the underlying distribution." is unclear.

Figure legend was extended to explain this point:

"The violin plot of all kinetic parameters from the fit results to a single cell calcium trajectory. The plot values are in terms of the log of 10 of the parameter values normalized according to the reference values determined either through literature or values within reasonable biological ranges. The meanings of the kinetic parameter symbols are listed in Table S1"

6. EV Fig.4

What does it mean that whiskers do not extend from the box? E.g. "eta1 (ER)" for WP.

What kind of box-plot is that? What do whiskers signify?

The confusion stems from a technical issue in figure export from Matlab to illustrator where whiskers had a dashed line. This was fixed.

7. Sentence, page 9, par. 1:

"To test if the nearest neighbours probability that we calculated shows a statistically significant difference..." -> difference

Corrected

8. Table S1 was missing from my copy of the manuscript.

Table S1 is now included.

9. Please explicitly define what "parameter vector" means.

We added the following definition to the methods section:

“A parameter vector is defined by the collection of parameters that are needed to fully specify the model shown in figure 2”

10. Since the model was simplified compared to the original, please deposit the model into BioModels database.

Model submission to BioModels database is in progress.

2nd Editorial Decision

11 November 2016

Thank you for sending us your revised manuscript. We have now heard back from reviewer #2 who agreed to evaluate the study. As you will see below, this reviewer thinks that the major issues have been satisfactorily addressed. However, s/he raises a remaining concern, which we would ask you to address in a minor revision.

REFEREE REPORT

Reviewer #2:

The clarity of the manuscript has improved significantly and the authors have addressed all of my concerns, except one.

At the beginning of page 4 they write:

"We initially attempted to cluster the single cell calcium trajectories based on either the entire calcium responses or through their representative time series features. However, no natural grouping of cells was observed using these two simple cell response signatures (Expanded View Figure 1)..."

I analysed the supplied data (74919_1_additional_figure_data_1020082_jfjt5j.mat) and obtained very good results with one of the most commonly used Ward's linkage method. Also k-means clustering (conceptually similar to Ward's) produces good clustering with Calinski criterion = 3. The results of my analysis along with R script to generate the plots is in the online folder:

<https://www.dropbox.com/sh/436hd2ptnoeodvv/AACLuFaVYN3nnDTrWUavWszva?dl=0>

The authors should therefore rewrite this paragraph and the caption of EV Fig. 1. In its current form it leaves a reader with an impression that the only way to classify such dataset would be through a model-based approach.

I am not questioning this approach, however. It certainly gives a valuable mechanistic insight into the source of heterogeneous responses as evidenced by clustering of model parameters and the experiment. However, the presentation should account for the fact that it is in fact possible to cluster experimental time courses. Additionally it would be interesting to compare such clustering with 3 clusters based on model parameters. Calinski criterion calculated from clustered time courses hints at the existence of 3 clusters, which is also the number of clusters calculated for model parameters.

Once the authors address this issue, I am happy to recommend it for publication.

Thank you for the opportunity to re-submit the paper after minor revisions. Below is point by point response to reviewer comments. For clarity we show reviewer comments in serif fonts italics and our response in san serif below.

Reviewer #2:

We thank the reviewer for his/her in depth analysis and for taking the time to reanalyze some of our data. The presented insights are meaningful and we are revising the paper as requested:

The authors should therefore rewrite this paragraph and the caption of EV Fig. 1. In its current form it leaves a reader with an impression that the only way to classify such dataset would be through a model-based approach.

We agree with reviewer #2 that cluster analysis is definitely possible based on timeseries data. To prevent any confusion and possible false impression that cluster based on timeseries is valid we edited the text as follows:

Main text:

We revised the statement that no natural clusters were found to the following:

The obtained clusters did not identify clear response archetypes (**Expanded View Figure 1**) potentially as a result of the fact that the ratio of variance between clusters to the intracellular variance was low (**Expanded View Figure 1d**). Therefore, while clustering based on timeseries data is a valid and useful approach, in this case it did not lead to identification of cell states. To further explore the possibility that cell states exist, we hypothesize that insights regarding cellular states could be uncovered by studying the mechanistic structure of pathway using mathematical modeling that incorporate prior knowledge on the underlying signaling network.

Legend of Expanded View Figure 1.

We removed the potentially confusing sentence: "The dendrogram shows that there was no clear clustering of the time course data."

Thank you again for sending us your revised manuscript. We are now satisfied with the modifications made and I am pleased to inform you that your paper has been accepted for publication.

Corresponding Author Name: Roy Wollman

Manuscript Number: MSB-16-7137